# Correlation induced electron-hole asymmetry in quasi- two-dimensional iridates

Ekaterina M. Pärschke[1], Krzysztof Wohlfeld[2], Kateryna Foyevtsova[3] & Jeroen van den Brink[1,4]

The resemblance of crystallographic and magnetic structures of the quasi-two-dimensional iridates $Ba_2IrO_4$ and $Sr_2IrO_4$ to $La_2CuO_4$ points at an analogy to cuprate high-Tc superconductors, even if spin-orbit coupling is very strong in iridates. Here we examine this analogy for the motion of a charge (hole or electron) added to the antiferromagnetic ground state. We show that correlation effects render the hole and electron case in iridates very different. An added electron forms a spin polaron, similar to the cuprates, but the situation of a removed electron is far more complex. Many-body $5d^4$ configurations form which can be singlet and triplet states of total angular momentum that strongly affect the hole motion. This not only has ramifications for the interpretation of (inverse–)photoemission experiments but also demonstrates that correlation physics renders electron- and hole-doped iridates fundamentally different.

[1] IFW Dresden, Helmholtzstr. 20, 01069 Dresden, Germany. [2] Institute of Theoretical Physics, Faculty of Physics, University of Warsaw, Pasteura 5, PL-02093 Warsaw, Poland. [3] University of British Columbia, 6224 Agricultural Road, Vancouver, BC, Canada V6T 1Z1. [4] Institute for Theoretical Physics, TU Dresden, 01069 Dresden, Germany. Correspondence and requests for materials should be addressed to E.M.P. (email: e.plotnikova@ifw-dresden.de)

Recently a large number of studies have been devoted to the peculiarities of the correlated physics found in the quasi-two-dimensional iridium oxides, such as e.g. $Sr_2IrO_4$ or $Ba_2IrO_4$[1–3]. It was shown that this $5d$ family of transition metal oxides has strong structural and electronic similarities to the famous $3d$ family of copper oxides, the quasi-two-dimensional undoped copper oxides as exemplified by $La_2CuO_4$ or $Sr_2CuO_2Cl_2$[1, 4, 5]. Moreover, just as for the cuprates, the ground state of these iridates is also a two-dimensional antiferromagnet (AF) and a Mott insulator[1, 5–7]—albeit formed by the $j = 1/2$ spin-orbital isospins instead of the $s = 1/2$ spins[1, 2, 5].

It is a well-known fact that the quasi-two-dimensional copper oxides turn into non-BCS superconductors when a sufficient amount of extra charge is introduced into their Mott insulating ground state[8]. On the basis of the above mentioned similarities between cuprates and iridates it is natural to ask the question[9] whether the quasi-two-dimensional iridates can also become superconducting upon charge doping. On the experimental side, very recently signatures of Fermi arcs and the pseudogap physics were found in the electron- and hole-doped iridates[7, 10–12] on top of the $d$-wave gap in the electron-doped iridate[11]. On the theoretical side, this requires studying a doped multiorbital two-dimensional Hubbard model supplemented by the non-negligible spin–orbit coupling[6, 13–18]. The latter is a difficult task, since even a far simpler version of this correlated model (the one-band Hubbard model) is not easily solvable on large, thermodynamically relevant, clusters[19].

Fortunately, there exists one nontrivial limit of the two-dimensional doped Hubbard-like problems, whose solution can be obtained in a relatively exact manner. It is the so-called single-hole problem which relates to the motion of a single charge (hole or doublon) added to the AF and insulating ground state of the undoped two-dimensional Hubbard-like model[20, 21]. In the case of the cuprates, this problem has been intensively studied both on the theoretical as well as the experimental side and its solution (the formation of the spin polaron) is considered a first step in understanding the motion of doped charge in the two-dimensional Hubbard model[22–25]. In the case of iridates several recent angle-resolved photoemission spectroscopy (ARPES) experiments unveiled the shape of the iridate spectral functions[1, 7, 11, 26–31]. However, on the theoretical side this correlated electron problem has not been investigated using the above approach[1, 6, 7, 32]—although it was suggested that the combination of local density approximation of density functional theory and dynamical mean field theory (LDA + DMFT) or even LDA + U band structure description might be sufficient[1, 27, 29, 30, 33, 34].

Here we calculate the spectral function of the correlated strong coupling model describing the motion of a single charge doped into the AF and insulating ground state of the quasi-two-dimensional iridate, using the self-consistent Born approximation (SCBA), which is well suited to the problem[22, 25, 35–38]. The main result is that we find a fundamental difference between the motion of a single electron or hole added to the undoped iridate. Whereas the single electron added to the $Ir^{4+}$ ion locally forms a $5d^6$ configuration, adding a hole (i.e., removing an electron) to the $Ir^{4+}$ ion leads to the $5d^4$ configuration. (We note here that in what follows we assume that the iridium oxides are in the Mott-Hubbard regime, since the on-site Hubbard $U$ on iridium is smaller than the iridium-oxygen charge transfer gap[14, 39, 40].) Due to the strong on-site Coulomb repulsion, these differences in the local ionic physics have tremendous consequences for the propagation of the doped electrons and holes. In particular, in the electron case the lack of internal degrees of freedom of the added charge, forming a $5d^6$ configuration, makes the problem qualitatively similar to the

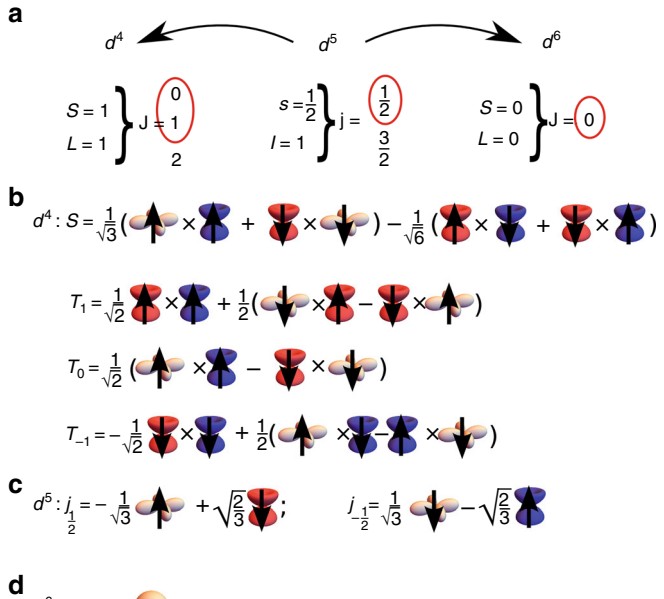

**Fig. 1** Low-energy eigenstates of iridium ions. **a** Quantum numbers characterising certain electronic configuration, where $j$, $l$, and $s$ ($J$, $L$, and $S$) stands for single-particle (multi-particle) total, orbital, and spin angular momentum. The red circles indicate the states that are explicitly taken into account in our effective low-energy theory. **b** Eigenstates for the $5d^4$ configuration (relevant for the $5d^4$ hole case) of the appropriate ionic Hamiltonian of iridium ion. **c** Same as **b** but for the $5d^5$ configuration (as relevant for the quasi-two-dimensional iridate ground state). **d** Same as **b** but for the $5d^6$ configuration (relevant for $5d^6$ doublon case). Blue, red and beige cartoon orbitals indicate the one-particle states with the effective angular momentum $l = 1$ and $l^z = 1$, $l^z = -1$ and $l^z = 0$ respectively. Round beige cartoon orbital indicates full shell with $L = 0$. Up (down) black arrows indicate $s^z = 1/2$ ($s^z = 1/2$) of spin $s = 1/2$ states. No arrow on an orbital indicates $S = 0$ state

above-discussed problem of the quasi-two-dimensional cuprates and to the formation of the spin polaron. On the contrary, the hopping of a hole to the nearest-neighbor site does not necessarily lead to the coupling to the magnetic excitations from $j = 1/2$ AF, which is a result of the fact that the $5d^4$ configuration may have a nonzero total angular momentum $J$[41]. As discussed in the following, this result has important consequences for our understanding of recent and future experiments of the quasi-two-dimensional iridates.

## Results

**Model.** We begin with the low-energy description of the quasi-two-dimensional iridates. In the ionic picture (i.e., taking into account in an appropriate "ionic Hamiltonain" the cubic crystal field splitting[42], the spin-orbit coupling[2], and the on-site Coulomb interaction[41]) the strong on-site spin-orbit coupling $\lambda$ splits the iridium ion $t_{2g}$ levels into the $j = 1/2$ lower-energy doublet (Fig. 1) and the $j = 3/2$ higher energy quartet, where $j$ is the isospin (total angular momentum) of the only hole in the $5d^5$ iridium shell[1, 2, 9, 43]. For the bulk, the strong on-site Hubbard repulsion between holes on iridium ions needs to be taken into account which leads to the localization of the iridium holes and the AF interaction between their $j = 1/2$ isospins in the two-dimensional iridium plane[2]. Consequently, this Mott insulating ground state possesses two-dimensional AF long-range order with low-energy excitations well described in the linear spin

wave approximation[44]

$$\mathcal{H}_{\text{mag}} = \sum_k \omega_k \left( \alpha_k^\dagger \alpha_k + \beta_k^\dagger \beta_k \right), \tag{1}$$

where $\omega_k$ is the dispersion of the (iso)magnons $|\alpha_k\rangle$ and $|\beta_k\rangle$, which depends on two exchange parameters $J_1$ and $J_2$ (Supplementary Note 1), and k is the crystal momentum. We note here that, although the size of the experimentally observed optical gap is not large (around 500 meV[40]), it is still more than twice larger than the top of the magnon band in the resonant inelastic x-ray scattering (RIXS) spectra (around 200 meV)[5, 43]. This, together with the fact that the linear spin wave theory very well describes the experimental RIXS spectra of the quasi-two-dimensional iridates[5, 43], justifies using the strong coupling approach.

Introducing a single electron into the quasi-two-dimensional iridates, as experimentally realised in an inverse photoemission (IPES) experiment, leads to the creation of a single $5d^6$ doublon in the bulk, leaving the nominal $5d^5$ configuration on all other iridium sites. Since the $t_{2g}$ shell is for the $5d^6$ configuration completely filled, the only eigenstate of the appropriate ionic Hamiltonian is the one carrying $J = 0$ total angular momentum. Therefore, just as in the cuprates, the $5d^6$ doublon formed in IPES has no internal degrees of freedom, i.e., $|d\rangle \equiv |J=0\rangle$, Fig. 1.

Turning on the hybridization between the iridium ions leads to the hopping of the $5d^6$ doublon between iridium sites $i$ and $j$: $\left| 5d_i^5 5d_j^6 \right\rangle \left\langle 5d_i^6 5d_j^5 \right|$. It is important to realize at this point that, although such hopping is restricted to the lowest Hubbard subband of the problem, it may change the AF configuration and excite magnons. In fact, magnons are excited during all nearest-neighbor hopping processes, since the kinetic energy conserves the total angular momentum. Altogether, we obtain the IPES Hamiltonian:

$$\mathcal{H}_{\text{IPES}} = \mathcal{H}_{\text{mag}} + \mathcal{H}_t^d, \tag{2}$$

where $\mathcal{H}_{\text{mag}}$ is defined above and the hopping of the single $5d^6$ doublon in the bulk follows from the spin-polaronic[21–23, 45] Hamiltonian

$$\mathcal{H}_t^d = \sum_k V_k^0 \left( d_{kA}^\dagger d_{kA} + d_{kB}^\dagger d_{kB} \right) + \sum_{k,q} V_{k,q} \left( d_{k-qB}^\dagger d_{kA} \alpha_q^\dagger + d_{k-qA}^\dagger d_{kB} \beta_q^\dagger + h.c. \right), \tag{3}$$

where A, B are two AF sublattices, the term $\propto V_k^0$ describes the next nearest and third neighbor hopping, which does not excite magnons (free hopping), and the term $\propto V_{k,q}$ describes the nearest neighbor coupling between the $5d^6$ doublon and the magnons as a result of the nearest neighbor electronic hopping (polaronic hopping, see above). While the derivation and exact expressions for $V$'s are given in the Supplementary Note 3, we remark here that they depend on the five hopping elements of the minimal tight-binding model: $t_1$ ($t'$, $t''$) describing nearest (next nearest, third) -neighbor hopping between the $d_{xy}$ orbitals in the $xy$ plane, $t_2$—the nearest neighbor in-plane hopping between the other two active orbitals, $d_{xz}(d_{yz})$, along the $x(y)$ direction, and $t_3$—the nearest neighbor hopping between $d_{xz}$ ($d_{yz}$) orbitals along the $y(x)$ direction. The values of these parameters ($t_1 = -0.2239$ eV, $t_2 = -0.373$ eV, $t' = -0.1154$ eV, $t'' = -0.0595$ eV, $t_3 = -0.0592$ eV) are found as a best fit of this restricted tight-binding model to the LDA band structure (Supplementary Note 2). While in what follows we use the above set of tight-binding parameters in the polaronic model, we stress that the final results are not critically sensitive to this particular choice of the model parameters.

Next, following similar logic we derive the microscopic model for a single hole introduced into the iridate, which resembles the case encountered in the photoemission (PES) experiment. In this case a single $5d^4$ hole is created in the bulk. Due to the strong Hund's coupling the lowest eigenstate of the appropriate ionic Hamiltonian for four $t_{2g}$ electrons has the total (effective) orbital momentum $L = 1$ and the total spin momentum $S = 1$[46]. Moreover, in the strong spin-orbit coupled regime the $L = 1$ and $S = 1$ moments the eigenstates of such an ionic Hamiltonian are the lowest lying $J = 0$ singlet $S$, and the higher lying $J = 1$ triplets $T_\sigma$ ($\sigma = -1, 0, 1$, split by energy $\lambda/2$ from the singlet state) and $J = 2$ quintets. Since the high-energy quintets are only marginally relevant to the low-energy description in strong on-site spin-orbit coupling $\lambda$[43] limit, one obtains[41] that, unlike e.g. in the cuprates, the $5d^4$ hole formed in PES is effectively left with four internal degrees of freedom, i.e., $|\mathbf{h}\rangle \equiv \{|S\rangle, |T_1\rangle, |T_0\rangle, |T_{-1}\rangle\}$, Fig. 1.

Once the hybridization between the iridium ions is turned on, the hopping of the $5d^4$ hole between iridium sites $i$ and $j$ is possible: $\left| 5d_i^5 5d_j^4 \right\rangle \left\langle 5d_i^4 5d_j^5 \right| = \left| 5d_i^5 \right\rangle \left\langle 5d_j^5 \right| \left| \mathbf{h}_j \right\rangle \left\langle \mathbf{h}_i \right|$. Similarly to the IPES case described above, in principle such hopping of the $5d^4$ hole may or may not couple to magnons. However, there is one crucial difference w.r.t. IPES: the $5d^4$ hole can carry finite angular momentum and thus the $5d^4$ doublon may move between the nearest-neighbor sites without coupling to magnons. Altogether, the PES Hamiltonian reads

$$\mathcal{H}_{\text{PES}} = \mathcal{H}_{\text{mag}} + \mathcal{H}_{\text{SOC}} + \mathcal{H}_t^h, \tag{4}$$

where $\mathcal{H}_{\text{SOC}} = \lambda/2 \sum_{k,\sigma=-1,0,1} T_{k\sigma}^\dagger T_{k\sigma}$ describes the on-site energy of the triplet states which follows from the on-site spin-orbit coupling $\lambda$ and the hopping of the single $5d^4$ hole in the bulk is described by the following spin-polaronic[20–22] Hamiltonian

$$\mathcal{H}_t^h = \sum_k \left( \mathbf{h}_{kA}^\dagger \hat{V}_k^0 \mathbf{h}_{kA} + \mathbf{h}_{kB}^\dagger \hat{V}_k^0 \mathbf{h}_{kB} \right) + \sum_{k,q} \left( h_{k-qB}^\dagger \hat{V}_{k,q}^\alpha h_{kB} \alpha_q^\dagger + h_{k-qA}^\dagger \hat{V}_{k,q}^\beta h_{kB} \beta_q^\dagger + h.c. \right), \tag{5}$$

where (as above) A,B are two AF sublattices, the term $\propto \hat{V}_k^0$ describes the nearest, next nearest, and third neighbor free hopping, and the terms $\propto \hat{V}_{k,q}^\alpha$ and $\propto \hat{V}_{k,q}^\beta$ describe the polaronic hopping. The detailed derivation and exact expressions for $\hat{V}$'s are given in Supplementary Note 4: while they again depend on the the five hopping parameters, we stress that their form is far more complex, and each $\hat{V}$ is actually a matrix with several nonzero entries.

**Numerical results.** Using the SCBA method[22, 25, 35, 36, 38] we calculate the relevant Green functions for: the single electron ($5d^6$ doublon, $|d\rangle$) doped into the AF ground state of the quasi-two-dimensional iridate: $G_{\text{IPES}}(k,\omega) = \langle \text{AF} | d_k \frac{1}{\omega - \mathcal{H}_{\text{IPES}} + i\delta} d_k^\dagger | \text{AF} \rangle$, and the single hole ($5d^4$ hole, $|\mathbf{h}\rangle$) doped into the AF ground state of the quasi-two-dimensional iridate: $G_{\text{PES}}(k,\omega) = \text{Tr} \langle \text{AF} | \mathbf{h}_k \frac{1}{\omega - \mathcal{H}_{\text{PES}} + i\delta} \mathbf{h}_k^\dagger | \text{AF} \rangle$. We note that using the SCBA method to treat the spin-polaronic problems is well-established and that the noncrossing approximation is well-justified[35, 36, 38]. We solve the SCBA equations on a finite lattice of $16 \times 16$ sites and calculate the imaginary parts of the above Green's functions—which (qualitatively) correspond to the theoretical IPES and PES spectral functions.

We first discuss the calculated angle-resolved IPES spectral function shown in Fig. 2b. One can see that the first addition state

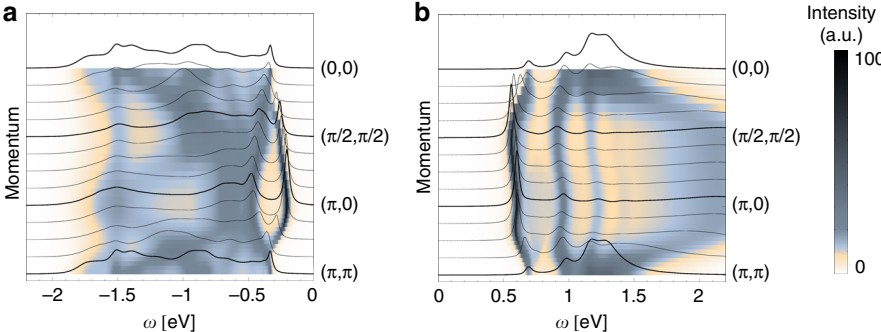

**Fig. 2** Theoretical spectral functions of iridates. **a** Photoemission (*PES*) and **b** inverse photoemission (*IPES*) spectral function of the low-energy (polaronic) models developed for the quasi-two-dimensional iridates and solved using the self-consistent Born approximation (see text). Following parameters are used: spin exchange $J_1 = 0.06$ eV, $J_2 = -0.02$ eV, $J_2 = 0.015$ eV, and spin-orbit coupling $\lambda = 0.382$ eV following ref. [43]; hopping integrals calculated as the best fit to the density functional theory (DFT) band structure (Supplementary Note 2) $t_1 = -0.2239$ eV, $t_2 = -0.373$ eV, $t' = -0.1154$ eV, $t_3 = -0.0592$ eV, $t'' = -0.0595$ eV; spectra offset by **a** $E = -0.77$ eV and **b** $E = -1.47$ eV; broadening $\delta = 0.01$ eV

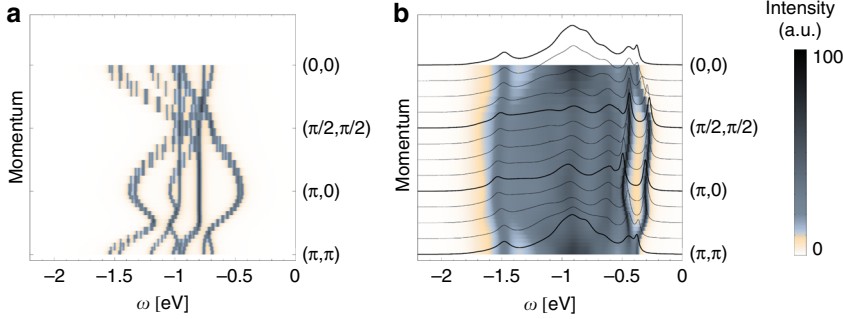

**Fig. 3** Free and polaronic contributions to the spectrum. **a** Theoretical photoemission spectral function with only propagation of the hole not coupled to magnons allowed as achieved by setting $\hat{V}_k^\alpha = \hat{V}_k^\beta \equiv 0$. **b** Theoretical photoemission spectral function with only polaronic propagation via coupling to magnons allowed (i.e., no free dispersion) as achieved by setting $\hat{V}_k^0 \equiv 0$. Parameters as in Fig. 2

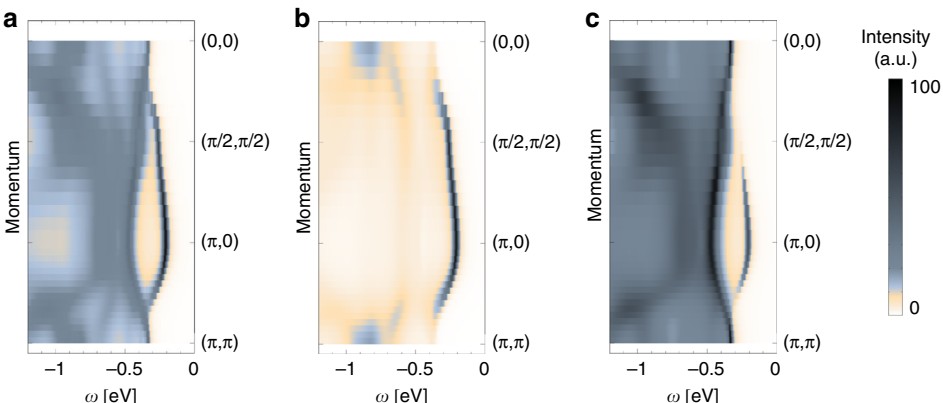

**Fig. 4** A zoom-in into the photoemission spectrum. **a** A zoom-in into the complete theoretical photoemission spectral function of Fig. 2a. **b**, **c** As **a** but $J$-resolved—with **b** showing the $J = 0$ contribution (motion of a "singlet hole") and **c** the $J = 1$ contribution (motion of a "triplet hole")

has a quasiparticle character, though its dispersion is relatively small (compared to the LDA bands, see Supplementary Note 2): there is a rather shallow minimum at $(\pi/2, \pi/2)$ and a maximum at the $\Gamma$ point. Moreover, a large part of the spectral weight is transferred from the quasiparticle to the higher lying ladder spectrum, due to the rather small ratio of the spin exchange constants and the electronic hopping[22]. Altogether, these are all well-known signatures of the spin-polaron physics: the mobile defect in an AF is strongly coupled to magnons (leading to the

ladder spectrum) and can move coherently as a quasiparticle only on the scale of the spin exchange $J_1$[20–22]. Thus, it is not striking that the calculated IPES spectrum of the iridates is similar to the PES spectrum of the $t$–$J$ model with a negative next nearest neighbor hopping—the model case of the hole-doped cuprates[23–25, 38]. This agrees with a more general conjecture, previously reported in the literature: the correspondence between the physics of the hole-doped cuprates and the electron-doped iridates[9].

Due to the internal spin and orbital angular momentum degrees of freedom of the $5d^4$ states, the angle-resolved PES spectrum of the iridates (Fig. 2a) is very different. The first removal state shows a quasiparticle character with a relatively small dispersion and a minimum is at the $(\pi, 0)$ point (so that we obtain an indirect gap for the quasi-two-dimensional iridates). On a qualitative level this quasiparticle dispersion resembles the situation found in the PES spectrum of the $t$–$J$ model with a positive next nearest neighbor hopping[25], which should model the electron-doped cuprates (or IPES on the undoped). However, the higher energy part of the PES spectrum of the iridates is quite distinct not only w.r.t. the IPES but also the PES spectrum of the $t$–$J$ model with the positive next nearest neighbor hopping[23–25]. Thus, the spin-polaron physics, as we know it from the cuprate studies[20–22], is modified in this case and we find only very partial agreement with the paradigm stating that the electron-doped cuprates and the hole-doped iridates show similar physics[9].

The above result follows from the interplay between the free (Fig. 3a) and polaronic hoppings (Fig. 3b) (we note that typically such interplay is highly nontrivial and the resulting full spectrum is never a simple superposition of these two types of hopping processes, cf. refs 23, 25, 47–50). The free hopping of the $5d^4$ hole is possible here for both the $J = 0$ singlet and $J = 1$ triplets which leads to the onset of several bands. As already stated, the $J = 1$ triplets can freely hop not only to the next nearest neighbors but also to the nearest neighbors (see above). For the polaronic hopping, the appearance of several polaronic channels, originating in the free $J$-bands being dressed by the $j = 1/2$ magnons, contributes to the strong quantitative differences w.r.t. the $5d^6$ doublon case or the cuprates.

**Comparison with experiment**. To directly compare our results with the experimental ARPES spectra of $Sr_2IrO_4$[1, 11, 27, 29], we plot the zoomed in spectra for PES, Fig. 4. Clearly, we find the first electron removal state is at a deep minimum at $(\pi, 0)$, in good agreement with experiment. This locus coincides with the $k$-point where the final state $J = 0$ singlet has maximum spectral weight, see Fig. 4b. Also the plateau around $(\pi/2, \pi/2)$ and the shallow minimum of the dispersion at the $\Gamma$ point are reproduced, where the latter is related to a strong back-bending of higher energy $J = 1$ triplets, see Fig. 4c. Thus one observes that the motion of the $5d^4$ hole with the singlet character is mostly visible around the minimum at $(\pi, 0)$ and near the plateau at $(\pi/2, \pi/2)$ (Fig. 4b), whereas the triplet is mostly visible at the $\Gamma$ points and much less at $(\pi, 0)$ (Fig. 4c). The higher energy features in the PES spectrum are mostly of triplet character, due to the difference in the on-site energies between the singlet and triplets $\propto \lambda$. These features, however, may in case of real materials be strongly affected by the onset of the oxygen states in the PES spectrum (not included in this study, see above).

Experimentally, electron doping causes Fermi arcs to appear in $Sr_2IrO_4$ that are centered around $(\pi/2, \pi/2)$[7, 10–12], which indeed corresponds the momentum at which our calculations place the lowest energy $d^6$ electron addition state. On the basis of the calculated electron-hole asymmetry one expects that for hole doping such Fermi arcs must instead be centered around $(\pi, 0)$, unless of course such doping disrupts the underlying host electronic structure of $Sr_2IrO_4$.

Finally, we note that, although the iridate spectral function calculated using LDA + DMFT is also in good agreement with the experimental ARPES spectrum[33], there are two well-visible spectral features that are observed experimentally, and seem to be better reproduced by the current study: the experimentally observed maximum at $\Gamma$ point in ARPES being 150–250 meV lower than the maximum at the $X$ point[1, 7, 11, 27–31], and the more

incoherent spectral weight just below the quasiparticle peak around the $\Gamma$ point than around the M point. We believe that the better agreement with the experiment of the spin polaronic approach than of the DMFT is due to the momentum independence of the DMFT self-energy—which means that the latter method is not able to fully capture the spin polaron physics[22, 51].

## Conclusions

The differences between the motion of the added hole and electron in the quasi-two-dimensional iridates have crucial consequences for our understanding of these compounds. The PES spectrum of the undoped quasi-two-dimensional iridates should be interpreted as showing the $J = 0$ and $J = 1$ bands dressed by $j = 1/2$ magnons and a free nearest and further neighbor dispersion. The IPES spectrum consists solely of a $J = 0$ band dressed by $j = 1/2$ magnons and a free next nearest and third neighbor dispersion. Thus, whereas the IPES spectrum of the quasi-two-dimensional iridates qualitatively resemble the PES spectrum of the cuprates, this is not the case of the iridate PES.

This result suggests that, unlike in the case of the cuprates, the differences between the electron and hole-doped quasi-two-dimensional iridates cannot be modeled by a mere change of sign in the next nearest hopping in the respective Hubbard or $t$–$J$ model. Any realistic model of the hole-doped iridates should instead include the onset of $J = 0$ and $J = 1$ quasiparticle states upon hole doping.

## Methods

**Derivation of the polaronic models**. The proper polaronic Hamiltonians, Eqs. (2) and (4), were derived from the DFT calculations and assuming strong on-site spin-orbit coupling and Coulomb repulsion. This was an analytic work which is described in details in Supplementary Notes 1–4 and which mostly amounts to: first, the downfolding of the DFT bands to the tight-binding (TB) model, second, the addition of the strong on-site spin-orbit coupling and Coulomb repulsion terms to the TB Hamiltonian, and thirdly the implementation of the successive: slave-fermion, Holstein-Primakoff, Fourier, and Bogoliubov transformations.

**Solving the polaronic models**. We calculated the respective Green's functions (see main text for details) for the polaronic model using SCBA. The SCBA is a well-established quasi-analytical method which, in the language of Feynman diagrams, can be understood as a summation of all so-called noncrossing Feynman diagrams of the polaronic model. It turns out that for the spin polaronic models (as, e.g., the ones discussed here) this approximate method works very well: the contribution of the diagrams with crossed bosonic propagators to the electronic Green's function can be easily neglected[35, 36, 38]. Although the SCBA method is in principle an analytical method, the resulting SCBA equations have to be solved numerically, in order to obtain results which can be compared with the experiment (such as, e.g., the spectral functions). The latter was done on a $16 \times 16$ square lattice (the finite size effects are negligible for a lattice of this size).

**Data availability**. The data supporting the present work are available from the corresponding author upon request.

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

## Acknowledgements

We thank Krzysztof Byczuk, Jiří Chaloupka, Dmitry Efremov, Marco Grioni, Andrzej M Oleś, Matthias Vojta, Robert Eder and Rajyavardhan Ray for stimulating discussions. K. W. acknowledges support by Narodowe Centrum Nauki (NCN, National Science Center) under Project No. 2012/04/A/ST3/00331 and Project No. 2016/22/E/ST3/00560. This work has been supported by the Deutsche Forschungsgemeinschaft via SFB 1143.

## Author contributions

E.M.P. derived the model and performed the SCBA calculations. K.F. performed the LDA calculations. J.v.d.B. and K.W. were responsible for project planning. K.W., E.M.P. and J.v.d.B. wrote the paper.

## Additional information

**Competing interests:** The authors declare no competing financial interests.

