## [Peer Review File · Nature Communications]

Reviewers' comments:

Reviewer #1 (Remarks to the Author):

The work is aimed at the single hole/electron dynamics in quasi-2D irridates. It is established that undoped irridates under discussion have the long range AF order. So the charge dynamics is determined by virtual emission/absorption of magnons. The paper employs the t-J-like model and the self-consistent Born approximation to calculate spectral functions. One of the most important conclusions is that the correlation physics in electron- and hole-doped irridates is fundamentally different.

This is timely and important work. The results and conclusions are interesting and important. However, I have two questions that have to be elucidated before I can recommend the paper.

1) The first question concerns justification of the t-J like model in the case of hole doping. It is known that in cuprates holes resides mostly on oxygen sites ($\sim 70\%$ of probability). Naively, this implies that the one band t-J model is not valid. Nevertheless the model is valid due to the Zhang-Rice singlet formation.

In irridates:

(i) What is the probability for hole to reside on oxygen sites?

This must be known from DFT calculations.

(ii) Is the Zhang-Rice-like physics applicable here too? It cannot be just singlet, there is also a triplet.

2) In the low energy sector the usual t-J model is equivalent to the Shraiman-Sigia model with fermions having pseudospin $1/2$ originating from two sublattices, magnons are described by non-linear sigma model. The fermion-magnon interaction is zero at $q \rightarrow 0$ according to Adler's theorem. Adler's theorem is reflected in Eq.(10) of the supplementary material, $V_{\{kq\}}$ is zero at $q=0$. The pseudospin $1/2$ is reflected in double degeneracy of spectral functions plotted in Fig2b.

Questions:

(i) In the case of hole in irridates, what is the degeneracy of spectral functions in Fig2a? In other words, what is the pseudospin of the hole?

(ii) Looking at Eqs.(19-24) describing the hole vertex one sees that some terms do not vanish at $q=0$. In principle this is possible. However, in the end Adler's thjeorem must be valid. Is the Adler's relation restored in the fully dressed hole Green's function?

Reviewer #2 (Remarks to the Author):

The authors present theoretical investigation on the many-body spectra function of the spin-orbit interaction mediated antiferromagnetic insulating states in iridates with square lattice. The main focus is the effect of spin-polarons, with detailed comparison to such excitations in cuprates.

The idea behind the manuscript, i.e., to study the correlations in the exotic $J=1/2$ states in iridates, is very interesting for both experimentalists and theorists. The methodology is well established, where detailed derivation has been specified for clear understanding. However, I would not recommend the manuscript as a publication on Nature Communications, because the arguments in the draft can only be qualitatively correct thus are descriptive.

(1) The authors selectively neglected the electronic structure of iridates obtained based on DMFT calculations, where a faithful treatment of local Coulomb interaction can be done. No doubt that the spin-polaron excitations can modify the physical properties significantly, but regarding the spectral function, it is not clear what is induced by such non-local excitations beyond DMFT.

For instance, following PRL 111, 246402 (13), the spectral function of Sr_2IrO_4 has been calculated, where all features claimed by the authors, such as indirect band gap, $J=0$ holes at $(\pi,0)$, $J=1$ triplet state at Γ , have already been obtained at the DMFT level considering local excitations.

In this regard, the authors are presenting biased arguments which requires more detailed clarification.

(2) I am not convinced by the tight-binding parameters used in the manuscript in order to compare with experimental ARPES measurements, as shown in Fig. 4. As shown in the supplementary information, the three parameters used can not fully account for the most important features of the LDA band structure. For instance, the down-turn of one band from $(\pi/2, \pi/2)$ to Γ across the Fermi energy indicates that hopping to further neighbors are required. For Fig. 4, yet two times larger tight-binding parameters are used, and it is not clear what the resulting non-interacting band structure looks like.

On the other hand, the fine features of PES and IPES the authors claim should be very sensitive to the starting electronic structure.

Therefore, even though the methods of the manuscript are well established and the idea behind is enlightening, the main conclusions about the non-local spin-polaron excitations are misleading and not convincing.

One minor comment is that the journal name for Ref. 10 is missing.

Reviewer #3 (Remarks to the Author):

In this paper, the authors have studied the motion of a charge (hole or electron) added to the proposed ground-state of quasi-2D iridates such as Ba_2IrO_4 or Sr_2IrO_4 . The authors show within the self-consistent Born approximation that the hole and electron carriers behave very differently. They claim that an added electron forms a spin-polaron, which resembles the well-known cuprates, but a removed electron is far more complex.

The authors' Hamiltonian is not at all clear neither in the d_6 case nor in the d_4 . How does the spin orbit interaction enter in the final Hamiltonian? Neither Eq.(10) nor Eq.(11) contain the parameter for the spin orbit interaction, because they depend only on t_1 , t_2 and t' which look the parameters

derived without the spin orbit interactions in (6) and (7) in the basis of xy , yz and zx orbitals.

After deriving the tight binding form including the contribution from the spin orbit interaction, the question is how one can be compatible between the small optical gap obtained in the optical conductivity (Moon et al. Phys. Rev. B 80, 195110 (2009)) and the strong coupling picture of the spin-wave analysis here.

Aside from them, the authors' claim that the PES excitation spectra are different from the electron doped PES is related to the higher energy bands away from the fermi level and looks details for understanding low energy excitations. The authors did not clarify what are the consequences of it.

Based on the above, I do not recommend this paper to publish in Nature Communications.

Reply to the Referees on the Manuscript NCOMMS-17-01445, Paerschke

May 11, 2017

1 Reply to Referee 1

1.1 Remark #1

The first question concerns justification of the t-J like model in the case of hole doping. It is known that in cuprates holes reside mostly on oxygen sites (70% of probability). Naively, this implies that the one band t-J model is not valid. Nevertheless the model is valid due to the Zhang-Rice singlet formation. In irridates: (i) What is the probability for hole to reside on oxygen sites? This must be known from DFT calculations. (ii) Is the Zhang-Rice-like physics applicable here too? It cannot be just singlet, there is also a triplet.

We are very grateful to the Referee for the above remark. Indeed the problem of the Zhang-Rice singlet formation might in principle be of crucial importance for the understanding of the properties of the doped irridates. However, there is considerable evidence that the irridium oxides do not lie in the charge transfer regime but instead should be regarded as the Mott-Hubbard insulators:

First, according to “our” LDA calculations the anti-bonding combinations of the strongly mixed Ir- t_{2g} orbitals and O- $2p$ orbitals of same symmetry are in the energy interval between -2 to 0.5 eV, and this is where the hole (in this uncorrelated picture) would go. However, unlike in the cuprates, the Ir- t_{2g} character dominates here meaning stronger localisation on the transition metal (Ir) sites compared to the cuprates. The relative contribution of the Ir- $5d$ is roughly 60 -70%.

Moreover, the *ab-initio* quantum chemical calculations suggest that the charge gap in the quasi-2D irridium oxides (Ba_2IrO_4 or Sr_2IrO_4) is of the order 0.5 eV, while the irridium-oxygen charge transfer gap is of the order of 2-3 eV, see Refs. [1]. There exists also strong evidence that the charge gap in the irridates is a Mott gap [2] that is much smaller than the charge transfer gap, putting the irridates in the Mott-Hubbard regime. Therefore the Zhang-Rice singlet physics should only affect the really high energy (2-3 eV) features of the PES spectrum which justifies neglecting the oxygens in this first correlated study of the PES / IPES spectra of the quasi-2D irridates.

Nevertheless, even though the Zhang-Rice singlet physics is most probably not of relevance for the relatively low energy features of PES that we consider here, we note that, as suggested in Ref. [1], interestingly, “a major difference

with respect to the 2D $S = 1/2$ cuprates is that in the iridates the lowest charge transfer states have apical hole character, whereas in the 2D Cu oxides the lowest electron-removal states have predominant in-plane O 2p character”.

We added the above remarks in the main text of the paper (see summary of changes).

1.2 Remark #2

In the low energy sector the usual t-J model is equivalent to the Shraiman-Sigia model with fermions having pseudospin 1/2 originating from two sublattices, magnons are described by non-linear sigma model. The fermion-magnon interaction is zero at $q = 0$ according to Adler’s theorem. Adler’s theorem is reflected in Eq.(10) of the supplementary material, V_{kq} is zero at $q = 0$. The pseudospin 1/2 is reflected in double degeneracy of spectral functions plotted in Fig2b. Questions: (i) In the case of hole in iridates, what is the degeneracy of spectral functions in Fig2a? In other words, what is the pseudospin of the hole? (ii) Looking at Eqs.(19-24) describing the hole vertex one sees that some terms do not vanish at $q=0$. In principle this is possible. However, in the end Adler’s theorem must be valid. Is the Adler’s relation restored in the fully dressed hole Green’s function?

We thank the Referee for these remarks. It can easily be shown that the PES spectra, just as the IPES spectra, are implicitly doubly degenerate - in this respect there is no difference between the standard spin 1/2 t-J model and our pseudospin 1/2 t-J model. The reason for this is that there are two ways to remove an electron from the d^5 shell, either removing it from the $j = 1/2, j_z = 1/2$ state or from the $j = 1/2, j_z = -1/2$ state. The pseudo-spin of the hole is thus 1/2. Also the point that the referee makes on Adler’s theorem is interesting - we checked that the self-energy vanishes identically for the lowest quasiparticle pole for both the PES and IPES part of the spectra in the self-consistent calculation. Thus Adler’s theorem is not violated. Vice versa, checking its validity explicitly in an analytical way is complicated by the self-energy having a rather complex matrix structure in our case.

We added the above remarks in the main text of the paper (see summary of changes).

2 Reply to Referee 2

2.1 Remark #1

The authors selectively neglected the electronic structure of iridates obtained based on DMFT calculations, where a faithful treatment of local Coulomb interaction can be done. No doubt that the spin-polaron excitations can modify the physical properties significantly, but regarding the spectral function, it is not clear what is induced by such non-local excitations beyond DMFT. For instance, following PRL 111, 246402 (13), the spectral function of Sr2IrO4 has been calculated, where all features claimed by the authors, such as indirect

band gap, $J=0$ holes at $(\pi,0)$, $J=1$ triplet state at Γ , have already been obtained at the DMFT level considering local excitations. In this regard, the authors are presenting biased arguments which requires more detailed clarification.

We are very grateful to the Referee for this very insightful remark. Let us firstly note that, while DMFT is a perfect method to understand the generic role of correlations, it is an approximate method which assumes a momentum-independent self-energy. Therefore, as already pointed by the Referee, the single-site DMFT cannot capture the spin polaron physics. For instance, the DMFT-like study of the problem of a single hole doped into the undoped ground state of the t - J model [3] does not allow for the existence quantum spin fluctuations and the formation of a mobile spin polaron. This fact is confirmed by the momentum-dependence of the self-energy of the spin polaron problem [4]. Similarly, in the here studied ‘spin-polaronic’ problem applicable to the iridates, the self-energy is also explicitly momentum dependent, as illustrated in Fig. 1 and Fig. 2, which makes the presented here result qualitatively distinct w.r.t. the one obtained using DMFT.

Naturally, the spin-polaronic approach to the problem, presented in the paper, is also an approximate method of calculating the quasi-2D iridate spectral function. For instance, the ligands are not taken into account in the model, the higher lying iridium multiplets are neglected, and the Trugman paths are not in the solution. Since these are completely different kinds of approximation than the ones used by the DMFT method, we believe that the only way to tell which method yields more ‘realistic results’ is to compare the results obtained using these methods with the experiment, Fig. 3. It turns out that, although the DMFT results indeed are in good agreement with the experimental ARPES spectrum, there are few important spectral features that are observed experimentally, are not well described by DMFT, but are present in our study [see Fig. 3]:

- Experimentally observed maximum at Γ point in ARPES is 150-250 meV lower than the maximum at the X point [5, 6, 7, 8, 9, 10, 11, 12].
- Far more incoherent spectral weight just below the quasiparticle peak around the Γ point than around the M point [9, 8, 12]
- Significant amount of spectral weight below the quasiparticle peak around the M point [9, 6, 12].

In order to make the above important remarks clear for the Reader, we modified the main text of the paper in appropriate places (see summary of changes).

2.2 Remark #2

I am not convinced by the tight-binding parameters used in the manuscript in order to compare with experimental ARPES measurements, as shown in Fig. 4. As shown in the supplementary information, the three parameters used can not fully account for the most important features of the LDA band structure. For instance, the

Figure 1: The momentum dependence of imaginary (a) and real (b) part of the self-energy calculated as the Greens function in the main text of the paper. Model parameters as in the main text of the paper: spin exchange $J_1 = 0.06$ eV, $J_2 = -0.02$ eV, $J_3 = 0.015$ eV, and spin-orbit coupling $\lambda = 0.382$ eV following Ref. [13]; tight-binding hopping integrals calculated as the best fit to the LDA bands [14] $t_1 = -0.2239$ eV, $t_2 = -0.373$ eV, $t' = -0.1154$ eV, $t_3 = -0.0592$ eV, $t'' = -0.0595$; spectra offset by (a) $E = -1.47$ eV and (b) $E = -0.77$ eV. Broadening $\delta = 0.01$ eV.

Figure 2: The imaginary and real part of the self-energy calculated using parameters as in the main text and Fig. 1 plotted for three high symmetry points in the Brillouin zone: (a) $(0,0)$, (b) $(\pi,0)$, and (c) $(\frac{\pi}{2}, \frac{\pi}{2})$.

down-turn of one band from $(\frac{\pi}{2}, \frac{\pi}{2})$ to Gamma across the Fermi energy indicates that hopping to further neighbors are required. For Fig. 4, yet two times larger tight-binding parameters are used, and it is not clear what the resulting non-interacting band structure looks like. On the other hand, the fine features of PES and IPES the authors claim should be very sensitive to the starting electronic structure. Therefore, even though the methods of the manuscript are well established and the idea behind is enlighting, the main conclusions about the non-local spin-polaron excitations are misleading and not convincing.

We are very thankful to the Referee for the comment on our choice of the tight-binding (TB) parameters. Indeed, once we extended the set of our TB parameters by adding the third neighbor hopping (t'') and allowing for a small but finite nearest neighbor hopping along the so-called “inactive axis” of the d_{xz} and d_{yz} orbitals in the xy plane (t_3), [see Eqs. (6)-(7) of the Supplementary Information], all of the important features of the LDA bands, including the down-turn of the band from the $(\pi/2, \pi/2)$ to the Γ point, are accounted for, see new Fig. 1 of the Supplementary Information. Moreover, using the derived-in-this-way TB parameters in our polaronic model, we obtained an even better

Figure 3: Theoretical (a) IPES and (b) PES spectral functions of quasi-2D iridates plotted in the same manner as the spectral functions calculated using DMFT in Ref. [15] using parameters as in the main text and Fig. 1.

agreement between the theoretical and the experimental result than for the previously used choice of the TB parameters.

We stress that the above mentioned polaronic calculations were now obtained *without* invoking the additional increase of the TB parameters in the polaronic model, as suggested in Ref. [16]. Therefore, taking also into account the fact that such an increase cannot be easily justified (see also discussion in Ref. [16]), we decided not include this correction anymore.

Concerning the fine-tuning of the TB parameters, let us note as follows: while of course the resulting PES or IPES spectra depend on the initial choice of the TB parameters, small changes in the TB parameters usually do not strongly affect the PES or IPES spectra (i.e. the obtained PES / IPES spectra are quite robust w.r.t. such changes). In particular, as long as the TB parameters provide a ‘relatively good’ fit to the LDA spectra, they lead to the SCBA spectra with all of the reported features. As an example we present here in the reply, the PES and IPES spectra calculated using a different set of TB parameters (called model TB2) that gives a slightly worse fit to the LDA band structure (see Fig. 4 below). One can see that although the parameters of Model TB2 are quite different than those of the model used in the main text of the paper, all of the main features of the PES or IPES spectra in the main text of the paper are also present in Fig 5 below.

In order to take into account the above results, we revised the manuscript

in appropriate places (see summary of changes).

Figure 4: Another set of the TB model parameters (model TB2): (a) A cartoon illustrating the t_1 , t_2 , t_3 , t' and t'' hopping paths between the Ir-5d- t_{2g} orbitals. (b) Comparison between the DFT (blue) and TB model TB2 (red) dispersion of the Ir-5d- t_{2g} bands in Sr_2IrO_4 . Hopping integrals are $t_1 = -0.186$ eV, $t_2 = -0.314$ eV, $t' = -0.1216$ eV, $t_3 = t'' = -0.0694$ eV. The Fermi energy is set to zero.

3 Reply to Referee 3

The authors' Hamiltonian is not at all clear neither in the $d6$ case nor in the $d4$. How does the spin orbit interaction enter in the final Hamiltonian? Neither Eq.(10) nor Eq.(11) contain the parameter for the spin orbit interaction, because they depend only on t_1 , t_2 and t' which look the parameters derived without the spin orbit interactions in (6) and (7) in the basis of xy , yz and zx orbitals.

We agree with the Referee that in the submitted version of the manuscript it was indeed not fully detailed how the spin-orbit coupling entered the final Hamiltonian. First of all we note that we assume that the spin-orbit coupling term is larger than exchange interaction J (typical assumption for iridates, see Refs. [5, 17, 18]), though smaller than the on-site Coulomb integrals which lead to the multiplet structure of the $5d^4$ configuration (see e.g. Ref. [16]) Naturally, we only consider t_{2g} shell since the cubic crystal field splitting being of order $2.5 - 4\text{eV}$ is far bigger than other scales in the problem [19]. Therefore, the ionic Hamiltonian $\mathcal{H}_{\text{ionic}} \simeq \mathcal{H}_{\text{SOC}} + \mathcal{H}_{\text{Coulomb}}$ (where $\mathcal{H}_{\text{SOC}} = \lambda \sum_{\mathbf{i}} \mathbf{l}_{\mathbf{i}} \mathbf{s}_{\mathbf{i}}$ and $\mathcal{H}_{\text{Coulomb}}$ describes the on-site Coulomb interactions, see e.g. Ref. [20]) is diagonal in the following ionic basis states (see Fig. 1 of the main text): (i) for the $5d^6$

Figure 5: Theoretical (a) PES and (b) IPES spectral functions for the quasi-2D iridates using the TB parameters from model TB2: hopping integrals are $t_1 = -0.186$ eV, $t_2 = -0.314$ eV, $t' = -0.1216$ eV, $t_3 = t'' = -0.0694$ eV; spectra offset by (a) $E = -0.69$ eV and (b) $E = -1.03$ eV. Broadening $\delta = 0.01$ eV.

configuration: one state with the total angular momentum $J = 0$ state, (ii) for the $5d^5$ configuration: two states with the total angular momentum $j = 1/2$, (iii) for the $5d^4$ configuration: three states with the total angular momentum $J = 1$ ('triplet') and one state with the total angular momentum $J = 0$ ('singlet') (higher lying multiplets are neglected in this low energy study, see Ref. [16]). In this basis, the spin-orbit coupling enters the final Hamiltonian [Eq. (2) or Eq. (4) of the main text] in three distinct ways:

1. Explicitly, as the local on-site term, which splits the 'triplets' from 'singlets' for the $5d^4$ configuration, see Eq. (4) of the main text and the text immediately below that equation.
2. Implicitly by leading to a particular form of \mathcal{H}_{mag} – this problem is discussed in detail in Ref. [17].
3. Implicitly by leading to a particular form of \mathcal{H}_t^d and \mathcal{H}_t^h , see below.

Presumably the main concern of the Referee is to understand how the spin-orbit coupling enters the Hamiltonian in the third case mentioned above, i.e. for \mathcal{H}_t^d and \mathcal{H}_t^h [given by Eqs. (7) and (10) of the SI]. We note that here, the spin-orbit coupling enters only in an implicit way, i.e. via the above mentioned form of the eigenstates $\mathcal{H}_{\text{ionic}}$: calculating the matrix elements of \mathcal{H}_t^d and \mathcal{H}_t^h in the polaronic basis basically boils down to calculating the matrix elements of the tight-binding Hamiltonian [Eq. (6) of the SI] in the appropriate eigenstates of $\mathcal{H}_{\text{ionic}}$.

In order to make the above point clear, we modified the manuscript in appropriate places (see summary of changes).

After deriving the tight binding form including the contribution from the spin orbit interaction, the question is how one can be com-

patible between the small optical gap obtained in the optical conductivity (Moon et al. Phys. Rev. B 80, 195110 (2009)) and the strong coupling picture of the spin-wave analysis here.

We are grateful to the Referee for this interesting remark on the strong coupling limit used here. We believe that using this approach is well justified here – the size of the optical gap is not large indeed, around 500 meV, but it is still more than twice larger than the largest energy of the magnon, at ca. 200 meV [see, for example, experimental data in [18, 13]]. Moreover, the linear spin wave theory very well describes the experimental RIXS spectra of the quasi-2D iridates and the broadening of the magnon excitations is lower than the currently available experimental resolution [18, 13].

We added a remark about this problem in the main text of the paper.

Aside from them, the authors’ claim that the PES excitation spectra are different from the electron doped PES is related to the higher energy bands away from the fermi level and looks details for understanding low energy excitations. The authors did not clarify what are the consequences of it.

Concerning the third remark, we point out that the claimed differences between calculated PES and IPES spectra are observed not only in the higher energy bands but also in the lowest energy part of the spectra, see page 6 and 7 of the main text. Thus, there is a rather shallow minimum at $(\pi/2, \pi/2)$ and a maximum at the Γ point in the IPES spectra whereas in the PES spectra the first electron removal state is at a deep minimum at $(\pi, 0)$ and there is a plateau around $(\pi/2, \pi/2)$ and the shallow minimum of the dispersion at the Γ point.

4 Summary of changes

(1) Following the advise of Referee 1 and Referee 3, we introduced a remark about the charge transfer and Zhang-Rice physics as applied to Iridates in the main text of the paper:

“We note here that in what follows we assume that the iridium oxides are in the Mott-Hubbard regime, since the on-site Hubbard U on iridium is smaller than the iridium-oxygen charge transfer gap [14, 38, 39].”

in lines 59-62 and added citations to the relevant papers, including Moon *et al.* [Phys. Rev. B 80, 195110 (2009)]. We also explicitly clarified the role of oxygens in our calculation in the following sentence in lines 213-215:

“These features, however, may in case of real materials be strongly affected by the onset of the oxygen states in the PES spectrum (not included in this study, see above)”

We also introduced the following remark on the size of the optical gap and strong coupling limit in lines 86-91

”We note here that, although the size of the experimentally observed optical gap is not large (around 500 meV [21]), it is still more than twice larger than the top of the magnon band in the RIXS spectra (around 200 meV) [18, 13]. This, together with the fact that the linear spin wave theory very well describes

the experimental RIXS spectra of the quasi-2D iridates [18, 13], justifies using the strong coupling approach.”

(2) In order to more clearly explain how the spin-orbit coupling entered the final Hamiltonian, as advised by Referee 3, we introduced the following clarification:

“(i.e. taking into account in an appropriate ‘ionic Hamiltonian’ the cubic crystal field splitting [19], the spin-orbit coupling λ [17], and the on-site Coulomb interaction [16])”

in lines 73-74. Also, we have modified the sentence about singlet and triplet formation in lines 131-134, so that now it reads:

“Moreover, in the strong spin-orbit coupled regime the $L = 1$ and $S = 1$ moments the eigenstates of such an ionic Hamiltonian are the lowest lying $J = 0$ singlet S , and the higher lying $J = 1$ triplets T_σ ($\sigma = -1, 0, 1$, split by energy λ from the singlet state) and $J = 2$ quintets.”

We have also extended and rewritten the first two paragraphs of section C of the Supplementary Information (lines 63-81) in order to make the derivation of our model more clear to the reader. After modification it reads

“Having obtained the TB Hamiltonian we are now ready to derive the Hamiltonian which would describe the motion of the “ $5d^6$ doublon” added to the Mott insulating ground state formed by the $5d^5$ iridium ions of the (undoped) quasi-2D iridates due to the nonzero hopping elements of the TB Hamiltonian. This means that the main task here is to calculate the following matrix elements of the tight-binding Hamiltonian [Eq. (6) above] $\langle 5d_1^6 5d_j^5 | \mathcal{H}_{\text{TB}} | 5d_1^5 5d_j^6 \rangle$. This is done in several steps:

Firstly, we calculate the above matrix elements in the appropriate eigenstates of ionic Hamiltonian of the $5d^5$ and $5d^6$ configurations (these states are listed in Fig. 1. of the main text). We note that these matrix elements do not explicitly depend on the strong on-site spin-orbit coupling λ , though the form of the appropriate eigenstates of the ionic Hamiltonian (Fig. 1 of the main text) is of course due to the onset of strong on-site spin-orbit coupling λ . Secondly, we assume the so-called no double occupancy constraint, which follows from the implicitly assumed here limit of strong on-site Coulomb repulsion – which prohibits the creation of “unnecessary” “ $5d^6$ doublons” once the electron added to the quasi-2D iridate $5d^5$ ground state hops between sites. Technically this amounts to the introduction of the projection operator which takes care of this constraint. Finally, following the path described for example in Refs. [4, 22] and introducing the slave-fermion formalism followed by Fourier and Bogoliubov transformations, we arrive at the following polaronic Hamiltonian which describes the motion of the “ $5d^6$ doublon”

(3) Motivated by insightful comments of Referee 2 on our choice of the tight-binding (TB) parameters we have extended set of our TB parameters and introduced the following changes into the new version of the paper:

1. The description of the extended TB model is given in lines 116-121
2. The sentence in lines 152-154 was modified accordingly, now being “the term $\propto \hat{V}_{\mathbf{k}}^0$ describes the nearest, next nearest, and third neighbor free hopping, and the terms $\propto \hat{V}_{\mathbf{k},\mathbf{q}}^\alpha$ and $\propto \hat{V}_{\mathbf{k},\mathbf{q}}^\beta$ describe the polaronic hopping”.

3. The sentence in lines 202-203 was changed to “To directly compare our results with the experimental ARPES spectra of Sr_2IrO_4 [5, 9, 6, 7], we plot the zoomed in spectra for PES, see Fig. 5.” to accommodate the change of the final TB parameters.
4. We have consequently modified Figures 2, 3 and 4.
5. To accomodate new figures, the caption of the Fig. 2 where the TB parameters are given was modified.
6. We have added the following comment concerning the fine-tuning of the TB parameters in lines 123-125:
 “While in what follows we use the above set of tight-binding parameters in the polaronic model, we stress that the final results are not critically sensitive to this particular choice of the model parameters.”
7. The Eq. (13)-(31) of the Supplementary Information that describe vertices were modified to comprehend new extended TB model.

(4) Following the advice of Referee 2 we modified the manuscript in order to relate our study to existing DMFT investigations, thus, we have modified the last sentence in the introduction (lines 49-52):

“However, on the theoretical side this correlated electron problem has not been investigated using the above approach [5, 23, 11, 24] - although it was suggested that the LDA+DMFT (or even LDA+U) band structure description might be sufficient [15, 5, 25, 6, 9, 10]“

and also added the following paragraph in lines 217-226:

“Finally, we note that, although the iridate spectral function calculated using LDA+DMFT is also in good agreement with the experimental ARPES spectrum [15], there are two well-visible spectral features that are observed experimentally, and seem to be better reproduced by the current study: (i) the experimentally observed maximum at Γ point in ARPES being 150-250 meV lower than the maximum at the X point [5, 6, 7, 8, 9, 10, 11, 12], and (ii) the more incoherent spectral weight just below the quasiparticle peak around the Γ point than around the M point. We believe that the better agreement with the experiment of the spin polaronic approach than of the DMFT is due to *inter alia* the momentum independence of the DMFT self-energy – which means that the latter method is not able to fully capture the spin polaron physics [4, 3].“

(5) Minor modifications of the text were made in several places in order to remove ambiguity, to provide additional explanation, to improve the reading, and to remove some misprints.

References

- [1] V. M. Katukuri, H. Stoll, J. van den Brink, and L. Hozoi, Phys. Rev. B **85**, 220402 (2012).
- [2] J.-M. Carter, V. Shankar, and H.-Y. Kee, Phys. Rev. B **88**, 035111 (2013).
- [3] R. Strack and D. Vollhardt, Phys. Rev. B **46**, 13852 (1992).

- [4] G. Martinez and P. Horsch, Phys. Rev. B **44**, 317 (1991).
- [5] B. J. Kim *et al.*, Phys. Rev. Lett. **101**, 076402 (2008).
- [6] A. de la Torre *et al.*, Phys. Rev. Lett. **115**, 176402 (2015).
- [7] Y. K. Kim, N. H. Sung, J. D. Denlinger, and B. J. Kim, Nat. Phys. **12**, 37 (2016).
- [8] Y. Liu *et al.*, Scientific Reports **5**, 13036 (2015).
- [9] Y. Nie *et al.*, Phys. Rev. Lett. **114**, 016401 (2015).
- [10] V. Brouet *et al.*, Phys. Rev. B **92**, 081117 (2015).
- [11] Y. Cao *et al.*, Nature Communications **7**, 11367 (2016).
- [12] A. Yamasaki *et al.*, Phys. Rev. B **94**, 115103 (2016).
- [13] J. Kim *et al.*, Nat. Commun. **5**, 4453 (2014).
- [14] See Supplementary Information for details.
- [15] H. Zhang, K. Haule, and D. Vanderbilt, Phys. Rev. Lett. **111**, 246402 (2013).
- [16] J. Chaloupka and G. Khaliullin, Phys. Rev. Lett. **116**, 017203 (2016).
- [17] G. Jackeli and G. Khaliullin, Phys. Rev. Lett **102**, 017205 (2009).
- [18] J. Kim *et al.*, Phys. Rev. Lett. **108**, 177003 (2012).
- [19] M. Moretti Sala *et al.*, Phys. Rev. B **90**, 085126 (2014).
- [20] A. M. Oleś, Phys. Rev. B **28**, 327 (1983).
- [21] S. J. Moon *et al.*, Phys. Rev. B **80**, 195110 (2009).
- [22] E. M. Plotnikova, M. Daghofer, J. van den Brink, and K. Wohlfeld, Phys. Rev. Lett. **116**, 106401 (2016).
- [23] H. Watanabe, T. Shirakawa, and S. Yunoki, Phys. Rev. B **89**, 165115 (2014).
- [24] B. H. Kim, T. Shirakawa, and S. Yunoki, Phys. Rev. Lett. **117**, 187201 (2016).
- [25] S. Moser *et al.*, New Journal of Physics **16**, 013008 (2014).

REVIEWERS' COMMENTS:

Reviewer #1 (Remarks to the Author):

I am fully satisfied by the revised manuscript and by the clarifications and changes made. The analysis of the spin-polaron physics for quasi-2D irridates, the account of correlations, and the demonstration of the hole-electron asymmetry is an important and interesting development. This is timely and important work, I recommend it for publication.

Reviewer #2 (Remarks to the Author):

In the revised version of the manuscript, the authors considered carefully my comments with proper correspondence. It is indeed very interesting to learn that the spin-polaron effects can be so strong that the fine features in the spectral functions are so well reproduced, somehow independent of the starting TB model parameters in the single particle picture. I am a bit conservative in this respect, though.

In short, the manuscript is well written with brilliant results seeking another possible way of understanding the $J=1/2$ insulating state in Sr_2IrO_4 . I would recommend it be accepted for publication in Nature Communications.